# Coagulation disorders in patients with severe hemophagocytic lymphohistiocytosis

**Sandrine Valade** [1,2]*, **Bérangère S. Joly**[2,3], **Agnès Veyradier**[2,3], **Jehane Fadlallah**[2,4], **Lara Zafrani**[1,2], **Virginie Lemiale**[1,2], **Amélie Launois**[2,3], **Alain Stepanian**[2,3], **Lionel Galicier**[2,4], **Claire Fieschi**[2,4], **Adrien Mirouse**[1,2], **Jean Jacques Tudesq**[1,2], **Anne-Claire Lepretre**[5], **Elie Azoulay**[1,2], **Michael Darmon**[1,2], **Eric Mariotte**[1,2]

**1** AP-HP, Medical ICU, Hôpital Saint-Louis, Paris, France, **2** EA3518, Université de Paris, Paris, France, **3** Hematology Biology Department, AP-HP, Hôpital Lariboisière, Paris, France, **4** Department of Clinical Immunology, AP-HP, Hôpital Saint-Louis, Paris, France, **5** Transfusion Department, Etablissement Français Du Sang, Hôpital Saint-Louis, Paris, France

* sandrine.valade@aphp.fr

**Data Availability Statement:** All relevant data are within the manuscript and its Supporting information files.

**Funding:** The author(s) received no specific funding for this work.

## Abstract

### Background

Coagulation disorders are common in patients with hemophagocytic lymphohistiocytosis (HLH), associated with an increased risk of bleeding and death. We aim to investigate coagulation disorders and their outcome implications in critically ill patients with HLH.

### Methods

We prospectively evaluated 47 critically ill patients with HLH (median age of 54 years [42–67]) between April 2015 and December 2018. Coagulation assessments were performed at day 1. Abnormal standard coagulation was defined as prothrombin time (PT) <50% and/or fibrinogen <2g/L. HLH aetiology was mostly ascribed to haematological malignancies (74% of patients).

### Results

Coagulation disorders and severe bleeding events were frequent, occurring in 30 (64%) and 11 (23%) patients respectively. At day 1, median fibrinogen level was 2·65g/L [1.61–5.66]. Fibrinolytic activity was high as suggested by increased median levels of D-dimers, fibrin monomers, PAI-1 (plasminogen activator inhibitor) and tPA (tissue plasminogen activator). Forty-one (91%) patients had a decreased ADAMTS13 activity (A Disintegrin-like And Metalloproteinase with ThromboSpondin type 1 repeats, member 13). By multivariable analysis, the occurrence of a severe bleeding (OR 3.215 [1.194–8.653], p = 0·021) and SOFA score (Sepsis-Related Organ Failure Assessment) at day 1 (OR 1.305 per point [1.146–1.485], p<0·001) were independently associated with hospital mortality. No early biological marker was associated with severe bleeding.

**Competing interests:** The authors have declared that no competing interests exist.

## Conclusions

Hyperfibrinolysis may be the primary mechanism responsible for hypofibrinogenemia and may also participate in ADAMTS13 degradation. Targeting the plasmin system appears as a promising approach in severe HLH-related coagulation disorders.

## Introduction

Hemophagocytic lymphohistiocytosis (HLH), or hemophagocytic syndrome, is a rare condition, represented by a severe systemic inflammatory state. The pathophysiology is most often supported by a deficient cytotoxicity in CD8 or NK lymphocytes resulting, after stimulation by a trigger, in an uncontrolled inflammatory response of macrophages [1–4]. This leads to high levels of circulating pro inflammatory cytokines, responsible for various biological abnormalities and clinical symptoms [5, 6]. HLH can be very severe and intensive care unit (ICU) admission is often required due to organ failures [7]. Prognosis of critically ill patients with HLH remains grim with high mortality rates, ranging between 40% and 80% [8–10]. Survival is especially poor in patients with underlying hematological malignancies [8, 9, 11–13].

Coagulation disorders are described in more than half of patients with HLH [9, 14, 15]. The most frequent reported abnormality is an isolated decrease in fibrinogen level [3, 6, 9, 14–18] whose mechanisms remain incompletely understood. Hypofibrinogenemia could be the result of primary fibrinolysis and/or disseminated intravascular coagulation (DIC), but no study has specifically focused on haemostasis pathways in HLH so far. There is also no data regarding involvement of primary haemostasis in HLH. Von Willebrand factor (VWF) is a multimeric glycoprotein essential for both platelet adhesion and aggregation after vascular injury. ADAMTS13 (A Disintegrin-like And Metalloproteinase with ThromboSpondin type 1 repeats, member 13) prevents the formation of platelet-rich thrombi by cleaving VWF ultralarge and hyperadhesive multimers. A severe deficiency of ADAMTS13 activity leads to thrombotic thrombocytopenic purpura (TTP), a specific thrombotic microangiopathy (TMA) characterized by neurologic and cardiac involvement [19–21]. However, despite its key role in primary haemostasis and its potential link with plasmin activation [22] ADAMTS13 involvement in HLH pathophysiology has never been investigated.

Hypofibrinogenemia is associated both with an increased risk of bleeding and a high mortality rate in earlier HLH studies [14–17, 23, 24]. However, only few data are available in critically ill HLH patients presenting with coagulation disorders. The largest study focusing on these patients reported that a fibrinogen level < 2 g/L was the only biological feature correlated with the occurrence of a severe haemorrhage [14]. Coagulation impairment is associated with an increased risk of death in HLH patients, especially low fibrinogen levels appear to be highly correlated with case fatality in retrospective studies [14–16]. So far, however, no clear mechanism leading to hypofibrinogenemia has been identified.

The main objective of this study was to explore in depth coagulation disorders in patients with severe HLH and to assess whether they are associated with bleedings and mortality. We also sought to identify early biomarkers associated with the occurrence of a bleeding event.

## Materials and methods

### Patients and blood collection

This prospective study was conducted in Saint Louis hospital (Paris, France) between April 2015 and December 2018. The Institutional Review Board (IRB 00006477v) of HUPNVS, Paris

7 University, AP-HP has approved the project (number 15–008). In accordance with the French legislation, the database was declared to the CNIL ("Commission Nationale de l'Informatique et des Libertés") (number 1837047v0). All adult patients diagnosed with HLH were included after written information and verbal consent. For specialized haemostasis investigation, venous blood was collected at day 1 into 1:10 final volume of 3.2% sodium citrate and double centrifuged (2500g for 10 minutes) to obtain platelet-poor plasma. Plasma samples were stored at -80˚C until tested.

## Definitions

HLH diagnosis was established according to the classification developed by the Histiocyte Society in 2004 (S1 Table) and was confirmed jointly by attending haematologists and intensivists. The HScore was obtained from clinical and biological data on day 1 (S2 Table). Aetiological diagnoses were made on a consensual basis, according to the results of diagnostic investigations. Based on a previous retrospective study from our group, coagulation disorders were defined by a prothrombin time (PT) < 50% and/or fibrinogen level < 2 g/L [14]. A severe haemorrhage consisted of a bleeding event requiring either red blood cells transfusion or haemostatic procedure (surgery or embolization), corresponding to grades 3–4 of the Common Terminology Criteria for Adverse Events (CTCAE v5) [25]. Organ failures were defined according to the Sepsis-Related Organ Failure Assessment (SOFA) score which was measured at admission [26]. Acute respiratory failure was defined by tachypnoea > 30/min, respiratory distress, SpO2 < 90% at ICU admission and/or laboured breathing [27]. Sepsis was established according to the 2001 task force definitions [28].

## Specialized haemostasis assays

All haemostasis assays were performed on platelet-poor plasma, according to the manufacturers' instructions.

First, fibrinolysis parameters were investigated. D-dimers (N <0.5 μg/mL) and fibrin monomers (N <6 μg/mL) were measured by immuno-turbidimetry using the STA-Liatest D-Di Plus® and the STA-Liatest FM® reagents, respectively, on the STA-R automate (Stago, Asnières-sur-Seine, France). Plasminogen concentration was measured (N: 80–120%) using the chromogenic STA-Stachrom® Plasminogen assay (Stago, Asnières-sur-Seine, France). Tissue plasminogen activator (t-PA) (N: 2–12 ng/mL) and plasminogen activator inhibitor-1 (PAI-1) (N: 4–43 ng/mL) were measured by ELISA using the Asserachrom tPA® and Asserachrom PAI-1® commercial kits, respectively (Stago, Asnières-sur-Seine, France).

Second, VWF and ADAMTS13 parameters were investigated. VWF antigen (Ag) (N: 50–150 IU/dL) was measured using the automated STA-Liatest VWF:Ag® (Diagnostica Stago, Asnières-sur-Seine, France). ADAMTS13 activity (N: 50–150 IU/dL) was measured with in-house FRETS-VWF73 assay using the recombinant VWF73 peptide (Peptide Institute, Osaka, Japan), as previously described [29]. ADAMTS13:Ag (N: 0·630–0·850 μg/mL) was measured using the Imubind® ADAMTS13 ELISA (BioMedica Diagnostics, Stamford, Connecticut, USA). Anti-ADAMTS13 IgGs (positivity threshold: 15 U/mL) were screened and titrated using the TECHNOZYM® ADAMTS-13 INH ELISA (Technoclone, Vienna, Austria).

## Statistical analysis

All quantitative variables were described using medians (quartiles) while qualitative variables were described by frequencies (percentage). Hospital mortality was the variable of primary interest. Correlations between biological and clinical characteristics were assessed using correlation matrix and correlation plots.

Independent predictors of mortality and severe bleeding were assessed using logistic regression models. Variables of interest were selected according to their relevance and statistical significance in univariate analysis. We used conditional forward stepwise regression with 0.2 as the critical P-value for entry into the model, and 0.1 as the P-value for removal. Interactions and correlations between the explanatory variables were carefully checked. Continuous variables for which log-linearity was not confirmed were transformed into categorical variables according to median or IQR. The final models were assessed by calibration, discrimination and relevancy. Residuals were plotted, and the distributions inspected.

Survivals were plotted using Kaplan Meier curves and compared using log-rank tests.

To assess influence of outliers on influence of fibrinogen on risk of severe hemorrhage, we used a bootstrapping technique, resampling the original set 1000 times with replacement then assessing distribution of fibrinogen in patients with and without severe hemorrhage and distribution of Odd ratios in a logistic regression with severe hemorrhage as variable of interest.

All tests were two-sided, and P-values less than 0.05 were considered statistically significant. Analyses were done using R software version 3.4.4 (https://www.r-project.org), including corrplot, survival, survminer and givitiR packages.

## Results

Overall, 47 patients (38% female) were included, median age 54 years [42–67], of whom 37 (79%) required ICU admission (Table 1). Critically ill patients were mainly admitted for acute respiratory distress (n = 14; 30%) or hemodynamic failure (n = 10; 21%). Almost two-thirds of patients (n = 29) were known immunocompromised at baseline (hematological malignancy, n = 13; HIV, n = 11; solid organ transplant, n = 2, other, n = 3). The patients fulfilled 5 [4–5] HLH 2004 criteria and median HScore was 244 [221–276]. Thirty nine (83%) of them presented with bicytopenia or pancytopenia and all except three had thrombocytopenia with a median platelet count of 47 x $10^9$/L [26–66]. Fever was almost constant and histological-cytological hemophagocytosis was found in 68% of the patients, mostly in bone marrow aspirate (n = 29). After an exhaustive investigative work up, HLH etiology was ascribed to haematological malignancy in 35 patients (74%, chiefly B-cell (n = 17) or T-cell lymphoma (n = 10)) (Table 1). Infectious disease was the HLH trigger in 7 patients (15%, represented by Mycobacteria- related infections in half of the cases), and auto-immune diseases triggered HLH in 2 patients. Three additional patients had an alternative diagnosis or an unknown etiology.

Thirty patients (64%) presented with coagulation disorders at day 1: median PT was 64% [48–72], median fibrinogen level was 2.65 g/L [1.61–5.66]. Regarding fibrinolytic activity in 45 patients with plasma samples available at day 1, median D-dimers and fibrin monomers levels were highly increased at 6.25 μg/mL [2.5–10] and 9 μg/mL [5–31], respectively (Fig 1). Although median plasminogen level was decreased at 57% [40–73], median levels of both tPA and PAI-1 were also increased at 45 ng/mL [31–67] and 94 ng/mL [45–188], respectively, (Fig 1). VWF antigen was highly elevated, mostly above the upper limit of quantification of the method (> 420 IU/dL), in all patients except one. ADAMTS13 antigen levels were slightly decreased at a median level of 0.264 μg/mL [0.149–0.371] (Fig 2). Interestingly, 41/45 (91%) of patients had a decreased ADAMTS13 activity (<50 IU/dL) and 20/45 patients (44%) had a severe functional deficiency in ADAMTS13 (activity <20 IU/dL). ADAMTS13 activity (median: 22 IU/dL [12–33]) was well correlated with ADAMTS13 antigen levels (median: 0.264 μg/mL [0.15–0.4]) (Fig 2). None of the 45 patients showed detectable anti-ADAMTS13 IgGs.

Eleven (23%) patients experienced a bleeding event; all of them were admitted to the ICU (Table 2). The most frequent localizations of haemorrhage were digestive tractus (n = 3) and

**Table 1. Characteristics of patients with hemophagocytic lymphohistiocytosis according to the outcome.**

| N (%) or Median [IQR] | Survivors (n = 29) | Non survivors (n = 18) | p |
|---|---|---|---|
| **Demographics** | | | |
| Age | 50 [35–67] | 55·5 [45·5–67] | 0·34 |
| Female gender | 12 (41%) | 6 (33%) | 0·81 |
| **Comorbidities** | | | |
| HIV infection | 7 (24%) | 4 (22%) | 1 |
| Hypertension | 9 (31%) | 6 (33%) | 1 |
| Diabetes | 3 (10%) | 1 (6%) | 0·97 |
| **ICU admission** | 20 (69%) | 17 (94%) | 0.09 |
| **Aetiological diagnosis** | | | |
| Onco-hematological malignancy | 21 (72%) | 14 (78%) | 0·95 |
| - B cell lymphoma | 11 | 5 | |
| - T or NK cell lymphoma | 6 | 5 | |
| - Hodgkin lymphoma | 3 | 3 | |
| - Other | 1 | 1 | |
| Infectious disease | 5 (17%) | 2 (11%) | 0·88 |
| Auto-immune disease | 2 (7%) | 0 | 0·69 |
| Unknown/alternative diagnosis | 1 (3%) | 2 (11%) | 0·67 |
| **HLH criteria** | | | |
| Hepatomegaly | 15 (52%) | 13 (72%) | 0·28 |
| Splenomegaly | 19 (66%) | 12 (67%) | 1 |
| Ferritin level (μg/L) | 11433 [5215–24474] | 18790 [8980–40347] | 0·19 |
| Triglycerides level (mmol/L) | 3·1 [1·9–3·9] | 3·05 [2·1–3·9] | 0·77 |
| Leucocytes (mm$^3$) | 2730 [1820–4470] | 6460 [1923–10268] | 0·29 |
| Hemoglobin (g/dL) | 8 [7·2–8·8] | 8·9 [7·4–9·1] | 0·22 |
| Platelets (x10$^9$/L) | 47 [28–66] | 38 [17–57] | 0·14 |
| Histological hemophagocytosis | 19 (66%) | 13 (72%) | 0·88 |
| HScore | 243 [219–269] | 249 [237–289] | 0·2 |
| **Hemostasis tests (day 1)** | | | |
| Prothrombin time (%) | 66 [55–80] | 51 [37–66] | **0·014** |
| Fibrinogen (g/L) | 2·93 [1·63–4·75] | 2·41 [1·41–5·67] | 0·71 |
| ADAMTS13 activity (IU/dL) | 25 [14–38] | 16 [11–28] | 0·14 |
| **SOFA score** | 4 [3–7] | 8.5 [6–12] | **<0·001** |
| **Severe hemorrhage** | 2 (7%) | 9 (50%) | **0·002** |
| **Treatments in the ICU** | | | |
| Mechanical ventilation | 4 (14%) | 11 (61%) | **0·002** |
| Vasopressors | 8 (28%) | 9 (50%) | 0·21 |
| Renal replacement therapy | 0 | 8 (44%) | **<0·001** |
| **HLH-related treatments** | | | |
| Etoposide | 19 (66%) | 15 (83%) | 0·32 |
| Corticosteroids | 19 (66%) | 16 (89%) | 0·15 |
| **Transfusion (day 1)** | | | |
| FFP (mL) | 0 | 412 [0–600] | **0·001** |
| Platelets (units) | 0 [0–6·2] | 7·4 [0–8] | **0·02** |

HIV, human immunodeficiency virus; ICU, intensive care unit; SOFA, Sepsis-related Organ Failure Assessment; FFP, fresh frozen plasma.

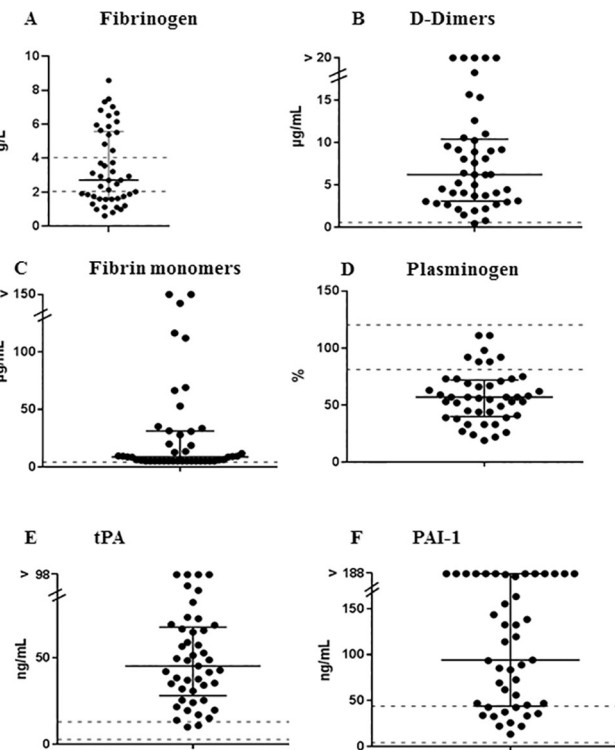

**Fig 1. Investigation for fibrinolysis-related parameters in 45 patients with HLH.** Each patient is represented by an icon. (A) fibrinogen; (B) D-dimers; (C) fibrin monomers; (D) plasminogen; (E) t-PA; (F) PAI-1.; normal ranges are represented as dashed lines; medians and IQR are represented as black lines.

puncture sites (n = 3), followed by intracranial (n = 2) and surgery sites (n = 2). Haemorrhage occurred 3 days [1–7] after HLH diagnosis and 2 days [0–5.5] after ICU admission. Median fibrinogen level was 1.46 g/L [1.29–2.64] at the onset of haemorrhage. Six patients required haemostatic procedures, either surgery (n = 3), endoscopy (n = 2) or vascular embolization (n = 1). Red blood cells (RBC) transfusions were used in 39 patients (83%), 34 patients (72%) received platelet transfusion and 19 (40%) received fresh frozen plasma (FFP). Five additional patients received fibrinogen concentrates. Median amount of transfused blood products was 3 RBC units [2–4], 27.2 platelets units [10–48.9] and 1650 mL of FFP [725–3300].

In the ICU, median SOFA score was 6 [3.5–9]. Fifteen patients (32%) required mechanical ventilation and 17 (36%) vasopressors. Eight patients underwent dialysis. Regarding HLH treatment, etoposide (VP16) was given to 72% of patients and corticosteroids to 74%. The majority of patients with haematological malignancies also received specific chemotherapy (n = 32, 91%).

Eighteen patients (38%) died during hospital stay. By univariate analysis, the occurrence of haemorrhage (p = 0.003), PT (p = 0·016), FFP transfusion (p = 0.002), platelets transfusion (p = 0.018), SOFA score (p < 0.001) and mechanical ventilation requirement (p = 0.003) were associated with hospital mortality (Table 2). Patients who experienced a bleeding event had more severe organ dysfunctions (median SOFA score 8 [5–12] versus 5 [3–9], p = 0.033; median lactate level 3.8 [3.6–8.2] versus 2.2 [1.4–3], p = 0.014), had a lower PT at day 1 (48% [37.5–60.5] versus 65% [53–79], p = 0.024), and received more often FFP (600 mL [0–110] versus none, p = 0.001). No significant difference was found regarding fibrinogen level between decedents and survivors (1.57 g/L [1.35–3.19] versus 2.94 g/L [1.69–5.6], p = 0.14).

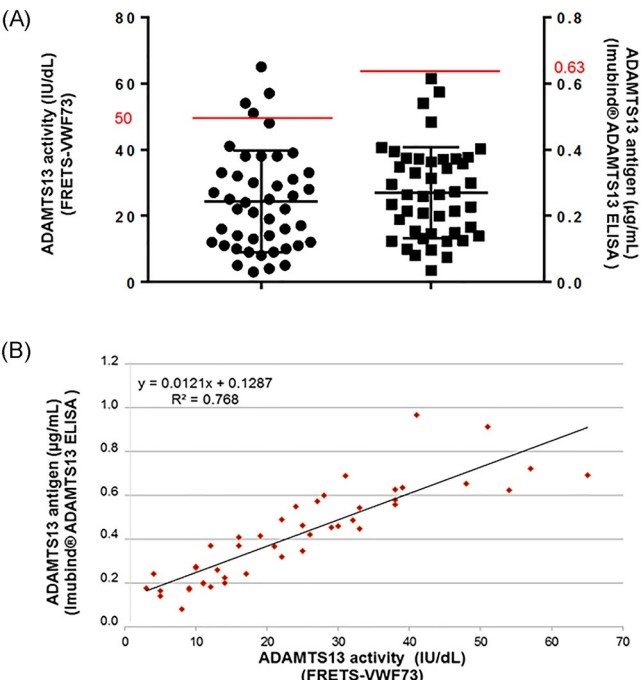

**Fig 2. Investigation for ADAMTS13 in 45 patients with HLH.** (A) Each patient is represented by an icon; ADAMTS13 activity and ADAMTS13 antigen levels are represented in the same graph; the upper limit of normal ranges are represented as red lines; medians and IQR are represented as black lines. (B) Correlation curve between ADAMTS13 activity and ADAMTS13 antigen.

Furthermore, we did not find any correlation across the different biological haemostasis parameters (S1 Fig).

By multivariable analysis, the occurrence of a severe haemorrhage (OR 3.2 [1.2–8.6], p = 0.02) and SOFA score (OR 1.3 per point [1.1–4.5], p < 0.001) were associated with hospital mortality (Fig 3). No specific haemostasis parameter was associated with bleeding events. In a post hoc analysis we assessed influence of outliers on fibrinogen distribution using bootstrapping technique. In the vast majority of resampled sets, fibrinogen level was higher in patients without haemorrhage than with haemorrhage (3.6 g/L [95%CI 3.5–3.6] vs. 2.8 g/L [95%CI 2.6–3.1]; P<0.001) suggesting a strong influence of outlier on the absence of difference in our dataset (S2 Fig).

## Discussion

To our knowledge, this is the first prospective study investigating haemostasis disorders, bleeding complications and assessing outcome in critically ill patients with HLH. Furthermore, we are able to propose pathophysiological hypotheses explaining the mechanisms of hypofibrinogenemia.

First, our results confirm that coagulation impairment is frequent during severe HLH, as almost two thirds of patients have PT < 50% and/or fibrinogen level < 2 g/L in this cohort. This is in line with previous studies in which up to 60% of patients presented haemostasis disorders [9, 14–18]. The association between coagulation disorders and prognosis in HLH patients has been previously reported. Particularly, a low fibrinogen level seems to be associated with adverse outcome [14–16], although we were not able to demonstrate a strong impact of any specific haemostasis parameters on mortality rate in this study, FFP transfusions being

**Table 2. Characteristics of patients with hemophagocytic lymphohistiocytosis according to the occurrence of a severe hemorrhage.**

| N (%) or Median [IQR] | Non-bleeding patients (n = 36) | Bleeding patients (n = 11) | p |
|---|---|---|---|
| **Demographics** | | | |
| Age | 52 [40–67] | 55·5 [45·5–69] | 0·39 |
| Female gender | 16 (44%) | 2 (18%) | 0·22 |
| **Comorbidities** | | | |
| HIV infection | 8 (22%) | 3 (27%) | 1 |
| Hypertension | 12 (33%) | 3 (27%) | 0·99 |
| Diabetes | 4 (11%) | 0 | 0·59 |
| **ICU admission** | 26 (72%) | 11 (100%) | 0·12 |
| **Aetiological diagnosis** | | | |
| Onco-hematological malignancy | 26 (72%) | 9 (82%) | 0·81 |
| - B cell lymphoma | 11 | 5 | |
| - T or NK cell lymphoma | 7 | 4 | |
| - Hodgkin lymphoma | 6 | 0 | |
| - Other | 2 | 0 | |
| Infectious disease | 6 (17%) | 1 (9%) | 0·89 |
| Auto-immune disease | 2 (6%) | 0 | 1 |
| Unknown/alternative diagnosis | 2 (6%) | 1 (9%) | 1 |
| **HLH criteria** | | | |
| Hepatomegaly | 22 (61%) | 6 (55%) | 0·97 |
| Splenomegaly | 24 (67%) | 7 (64%) | 1 |
| Ferritin level (µg/L) | 10757 [5660–21092] | 32489 [12273–42317] | 0·053 |
| Triglycerides level (mmol/L) | 2·85 [1·87–3·78] | 3·6 [3·0–4·4] | 0·15 |
| Leucocytes (mm$^3$) | 2835 [1795–7773] | 8660 [2035–10125] | 0·33 |
| Hemoglobin (g/dL) | 8·0 [7·2–8·8] | 9·1 [8·5–9·5] | 0·02 |
| Platelets (x10$^9$/L) | 47 [25–64] | 39 [24–56] | 0·56 |
| Histological hemophagocytosis | 25 (69%) | 7 (64%) | 1·00 |
| HScore | 246 [220–272] | 244 [232–287] | 0·57 |
| **Hemostasis tests (day 1)** | | | |
| Prothrombin time (%) | 66 [53–79] | 48 [38–61] | **0·024** |
| Fibrinogen (g/L) | 2.94 [1.69–5.6] | 1.57 [1.35–3.19] | 0·14 |
| ADAMTS13 activity (IU/dL) | 20 [11–32] | 27 [19–33] | 0·309 |
| **SOFA score** | 5 [3–9] | 8 [5·5–12] | **0·033** |
| **Treatments in the ICU** | | | |
| Mechanical ventilation | 9 (25%) | 6 (55%) | 0·14 |
| Vasopressors | 10 (28%) | 7 (64%) | 0·07 |
| Renal replacement therapy | 4 (11%) | 4 (36%) | 0·14 |
| **HLH-related treatments** | | | |
| Etoposide | 25 (69%) | 9 (82%) | 0·68 |
| Corticosteroids | 24 (67%) | 11 (100%) | 0·07 |
| **Transfusion (day 1)** | | | |
| FFP (mL) | 0 | 600 [0–1100] | **0·001** |
| Platelets (units) | 0 | 5·8 [0–8] | 0·25 |
| **Hospital death** | 9 (25%) | 9 (82%) | **0·003** |

HIV, human immunodeficiency virus; ICU, intensive care unit; SOFA, Sepsis-related Organ Failure Assessment; FFP, fresh frozen plasma.

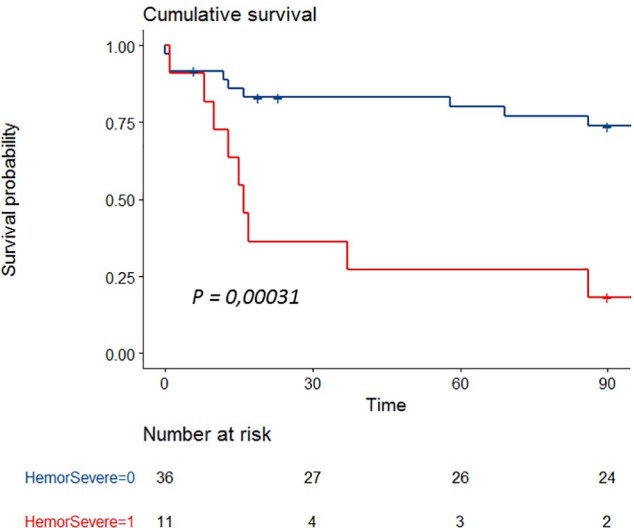

**Fig 3. Survival curve according to the occurrence of a severe haemorrhage.**

a major confounding factor. The relationship between haemostasis disorders and prognosis remains unclear and may be linked to the occurrence of bleeding complications. Few studies have specifically focused on haemorrhages in small series of HLH patients [14, 17, 24]. In our study, 23% of patients experienced a bleeding event, which strengthens the results of our previous retrospective study in which bleeding complications occurred in one fifth patients [14]. Timing of haemorrhage is early, arising three days after HLH diagnosis. Moreover, we demonstrated that the occurrence of haemorrhage was strongly associated with mortality: 82% of patients who experienced a bleeding complication died, compared to 26% of those who did not. We also confirmed the association between coagulation disorders and the occurrence of bleeding events, even if PT value was the only haemostasis parameter significantly associated with haemorrhage in this study. Fibrinogen level was decreased in bleeding patients (median 1.57 g/L) without reaching the threshold of significance. In the literature, several studies support that fibrinogen level is associated with the occurrence of severe haemorrhages in HLH patients, with various cut-off values between 1.5 and 2 g/L.

This last point highlights the need to explore the mechanisms leading to hypofibrinogenemia in HLH process. We have here investigated for the first time the haemostasis pathway in critically ill HLH patients and we have obtained some arguments supporting that hypofibrinogenemia may be mostly related to primary hyperfibrinolysis and not DIC. First, we found that PAI-1 and tPA levels were elevated in our patients, suggesting increased levels of plasmin, the predominant enzyme responsible for fibrinolysis. This could be in line with in vitro studies that have shown activated macrophages can release plasminogen activator [30]. Recently, data obtained from a murine model of fulminant HLH [31] showed an increase in tPA and plasmin-antiplasmin complex levels, indicating that the fibrinolytic system was over-activated during HLH. More interestingly, plasmin inhibition leads to a decrease in fibrinogen degradation products levels, attenuates pro-inflammatory cytokines production, reduces macrophages recruitment and improves survival in HLH-mice [31]. Plasmin appears to have a key role in HLH process, not only by generating haemostasis disorders through a decrease in fibrinogen level, but as a major actor of the inflammatory response. This is also supported by our previous retrospective study [14] showing that HLH patients with coagulation disorders had a more intense hemophagocytic activity with higher ferritin levels, more frequent features of

hemophagocytosis on bone marrow examination and also presented with more organ failures. Further specific studies are warranted in order to specify plasmin involvement in HLH-related coagulation disorders, and to define whether plasmin could be a potential therapeutic target in HLH.

We also interestingly demonstrated that more than 40% of patients had ADAMTS13 activity < 20 IU/dL, without any associated clinical or biological features of TMA. To our knowledge, no previous study has ever explored ADAMTS13 involvement in HLH. During sepsis and trauma, several authors have reported a reduction of ADAMTS13 activity in plasma and its association with disease severity and outcome [32, 33]. In a prospective study including 72 patients with septic shock, Peigne et al demonstrated that half patients had decreased ADAMTS13 activity < 30 IU/dL. This partial ADAMTS13 functional deficiency was likely related to an inhibition of its catalytic site by high levels of IL-6 rather than any degradation by DIC-related enzymes, as ADAMTS13 activity and ADAMTS13 antigen levels were not correlated [34]. Contrary to Peigne *et al*, we found a very good correlation between ADAMTS13 activity and ADAMTS13 antigen, which supports a quantitative defect in relation with protein degradation or synthesis deficiency, rather than an immune antibodies- or an inflammatory cytokines-mediated mechanism. Additional hypotheses can be suggested to explain ADAMTS13 quantitative deficiency. Firstly, ADAMTS13 may be consumed by the very high levels of its substrate VWF released from the inflammatory activated endothelial cells. Secondly, the macrophage haemoglobin scavenger receptor CD163, whose expression is restricted to the monocyte-macrophage lineage, has been found as a potential marker of HLH in humans [35, 36]. Its expression is correlated with enhanced phagocytic activity [37] and its extracellular part is cleaved upon macrophages activation, leading in high soluble CD163 levels in HLH patients. More recently, Verbij *et al* demonstrated in vitro that ADAMTS13 undergoes endocytosis by CD163-expressing macrophages [38]. This mechanism could explain the decrease in ADAMTS13 in HLH, taking into account the intense macrophages activation. Thirdly, several *in vitro* data suggest that ADAMTS13 activity could be impaired after undergoing proteolytic inactivation by plasmin and thrombin [39, 40]. Indeed, Crawley *et al* demonstrated that ADAMTS13 was rapidly cleaved by exogenous plasmin at low concentrations. This proteolytic inactivation results in the loss of ADAMTS13 activity. In HLH, as we previously demonstrated that our data support fibrinolytic pathway activation, we can therefore suppose that ADAMTS13 may be inactivated during hyperfibrinolytic state through high levels of circulating plasmin. This reinforces the fact that one of the main mechanisms responsible of haemostasis disorders in HLH could be in relation with primary hyperfibrinolysis.

However this study has some limitations. First, the small number of patients certainly has underpowered the analysis, especially regarding the impact of fibrinogen level which is supported by the bootstrap analysis. However HLH remains infrequent and few prospective studies have been conducted in the ICU. Second, due to its single-center design and our hospital specificity, a majority of patients with hematological malignancies have been included, even though haemostasis disorders have also been described in HLH patients with infectious or autoimmune triggers. Third, fourteen patients (30%) have received FFP transfusion at day 1, which could have overestimated ADAMTS13 and fibrinogen levels. Then, only preliminary data at day 1 have been analyzed; a future analysis including all haemostasis parameters during the first week monitoring is expected.

## Conclusions

This study is the first prospective one specifically focusing on coagulation impairment in severe HLH patients. Haemostasis disorders are common in critically ill patients with HLH

and are responsible of severe haemorrhages. Bleeding complications occur in nearly 25% of patients with an early timing and are associated with a high mortality rate. Several data suggest that hypofibrinogenemia may be the result of hyperfibrinolysis. This hypothesis is also supported by a decrease in ADAMTS13 activity, in the absence of TTP features, which is a new emerging concept. Further investigation of haemostasis parameters is warranted to clarify the role of plasmin, in order to identify new potential therapeutic targets in HLH.

## Supporting information

**S1 Table. HLH 2004 criteria (adapted from Henter et al, Pediatr Blood Cancer 2007).** (DOCX)

**S2 Table. HScore (adapted from Fardet et al, Arthritis Rheumatol 2014).** (DOCX)

**S1 Fig. Matrix of correlation between biological hemostasis parameters.** (TIF)

**S2 Fig. Fibrinogen distribution after bootstrapping.** (TIF)

## Acknowledgments

The authors would like to thank Sandrine Benghezal, Sylvaine Savigny and Sophie Capdenat for expert technical assistance. The authors thank Dr Issa Kalidi for the collection of plasma samples.

## Author Contributions

**Conceptualization:** Sandrine Valade, Bérangère S. Joly, Agnès Veyradier, Alain Stepanian, Lionel Galicier, Elie Azoulay, Eric Mariotte.

**Data curation:** Sandrine Valade, Jehane Fadlallah, Lara Zafrani, Virginie Lemiale, Amélie Launois, Alain Stepanian, Lionel Galicier, Claire Fieschi, Adrien Mirouse, Jean Jacques Tudesq, Anne-Claire Lepretre, Eric Mariotte.

**Formal analysis:** Michael Darmon.

**Investigation:** Sandrine Valade.

**Methodology:** Sandrine Valade, Bérangère S. Joly, Agnès Veyradier, Amélie Launois, Alain Stepanian, Elie Azoulay, Eric Mariotte.

**Supervision:** Agnès Veyradier, Elie Azoulay, Eric Mariotte.

**Validation:** Eric Mariotte.

**Writing – original draft:** Sandrine Valade.

**Writing – review & editing:** Bérangère S. Joly, Agnès Veyradier, Lionel Galicier, Elie Azoulay, Eric Mariotte.

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
