## [Decision Letter · Decision Letter 0]

12 Jul 2021

Coagulation disorders in patients with severe hemophagocytic lymphohistiocytosis

PONE-D-21-12544

Dear Dr. Valade,

We’re pleased to inform you that your manuscript has been judged scientifically suitable for publication and will be formally accepted for publication once it meets all outstanding technical requirements.

Kind regards,

Wolfgang Miesbach, MD

Academic Editor

PLOS ONE

Academic Editor's comment: Accept

Reviewers' comments:

Reviewer's Responses to Questions

**Comments to the Author**

1. Is the manuscript technically sound, and do the data support the conclusions?

Reviewer #1: Yes

2. Has the statistical analysis been performed appropriately and rigorously? 

Reviewer #1: Yes

3. Have the authors made all data underlying the findings in their manuscript fully available?

Reviewer #1: Yes

4. Is the manuscript presented in an intelligible fashion and written in standard English?

Reviewer #1: Yes

5. Review Comments to the Author

Reviewer #1: Thank you for your article titled, Coagulation disorders in patients with severe hemophagocytic lymphohistiocytosis' for consideration of publication in PLOS One Journal. HLH is a rare but severe life threatening disorder. Authors point out to an important issue of coagulation abnormalities in severe HLH, leading to clinically significant bleeding events. Also, bleeding is related to poor outcomes. Hopefully, with this article, clinicians can focus on coagulation abnormalities earlier and try to correct them fast, so that bleeding events are minimized. It should be accepted for publication without any revision needed.

6. PLOS authors have the option to publish the peer review history of their article (what does this mean?). If published, this will include your full peer review and any attached files.

Reviewer #1: **Yes: **Sumit Kapoor, MD

---

## [Editor Report · Acceptance letter]

26 Jul 2021

PONE-D-21-12544 

Coagulation disorders in patients with severe hemophagocytic lymphohistiocytosis 

Dear Dr. Valade:

I'm pleased to inform you that your manuscript has been deemed suitable for publication in PLOS ONE. Congratulations! Your manuscript is now with our production department. 

Kind regards, 

on behalf of

Dr. Wolfgang Miesbach 

Academic Editor

PLOS ONE